

# Short reads from honey bee (*Apis* sp.) sequencing projects reflect microbial associate diversity

Michael Gerth and Gregory D.D. Hurst

Institute of Integrative Biology, University of Liverpool, Liverpool, United Kingdom

## ABSTRACT

High throughput (or 'next generation') sequencing has transformed most areas of biological research and is now a standard method that underpins empirical study of organismal biology, and (through comparison of genomes), reveals patterns of evolution. For projects focused on animals, these sequencing methods do not discriminate between the primary target of sequencing (the animal genome) and 'contaminating' material, such as associated microbes. A common first step is to filter out these contaminants to allow better assembly of the animal genome or transcriptome. Here, we aimed to assess if these 'contaminations' provide information with regard to biologically important microorganisms associated with the individual. To achieve this, we examined whether the short read data from *Apis* retrieved elements of its well established microbiome. To this end, we screened almost 1,000 short read libraries of honey bee (*Apis* sp.) DNA sequencing project for the presence of microbial sequences, and find sequences from known honey bee microbial associates in at least 11% of them. Further to this, we screened ∼500 *Apis* RNA sequencing libraries for evidence of viral infections, which were found to be present in about half of them. We then used the data to reconstruct draft genomes of three *Apis* associated bacteria, as well as several viral strains *de novo*. We conclude that 'contamination' in short read sequencing libraries can provide useful genomic information on microbial taxa known to be associated with the target organisms, and may even lead to the discovery of novel associations. Finally, we demonstrate that RNAseq samples from experiments commonly carry uneven viral loads across libraries. We note variation in viral presence and load may be a confounding feature of differential gene expression analyses, and as such it should be incorporated as a random factor in analyses.

**Submitted** 4 March 2017
**Accepted** 11 June 2017
**Published** 12 July 2017

Corresponding author
Michael Gerth, gerth@liverpool.ac.uk

## INTRODUCTION

Novel DNA sequencing methods have revolutionized biological and medical research in the last two decades (*Goodwin, McPherson & McCombie, 2016*). High throughput sequencing (or 'massively parallelized sequencing', 'next generation sequencing','NGS') facilitated the creation of enormous amounts of data for a fraction of the costs associated with traditional Sanger methods (*Kircher & Kelso, 2010*; *Sboner et al., 2011*). This 'genomics revolution', has not only enhanced our understanding of molecular and genome evolution (*Wolfe &*

*Li, 2003*), but also contributed to the recognition that eukaryotes are commonly associated with a plethora of microbial taxa. (For the sake of simplicity, we will in this paper refer to all bacteria, microbial eukaryotes, and viruses as 'microbes'.)

In eukaryote genome sequencing projects, sequences deriving from these microbes may obstruct genome or transcriptome assembly efforts, and measures directed at removing microbial associates are routinely performed. This is achieved either by antibiotic treatment of the target organism prior to sequencing (*Colbourne et al., 2011*), or by removing microbial sequences bioinformatically after sequencing (*Schmieder & Edwards, 2011*). While eliminating microbes may facilitate eukaryotic genome reconstruction, it neglects the recently emerging appreciation of microbes as a biologically important component of all multicellular life. Numerous examples illustrate the impact of microbes on animal and plant biology, including physiology, behavior, and evolution (*McFall-Ngai et al., 2013*). These findings have led to a concept that defines an individual eukaryote with all its associated microbes (microbiome) as an entity (holobiont-hologenome) (*Bordenstein & Theis, 2015*). Although this concept is contentious (*Moran & Sloan, 2015*; *Douglas & Werren, 2016*), it is undisputed that some aspects of organismal biology can only be understood by deciphering interactions with microbial symbionts.

To characterize microbiome composition, three approaches are commonly used. First, microbes may be isolated from the host and cultured axenically. Their properties can then be determined either through traditional microbiological methods or by sequencing (*Browne et al., 2016*). This approach has the benefit of providing both biological and genomic information, but limits discovery to culturable taxa. Second, microbiome taxa may be identified by amplicon sequencing. Specific primers are used to amplify a short informative region from all bacterial taxa in a sample (usually a part of the 16S rRNA gene), and then sequenced (today typically via NGS methods) (*Caporaso et al., 2012*). This mechanism discovers broad patterns of community diversity, but at a coarse scale, and with weaker functional information. Finally, microbiome composition can be determined via metagenomics, i.e., collective genome sequencing of all bacteria present in a sample (*Riesenfeld, Schloss & Handelsman, 2004*). This is unbiased, fine scaled, and provides an assessment of biological potential at a community scale. Approaches to characterize viral sequences found in the environment or hosts are similar to the ones used to detect bacterial microbes described above and metagenomics is now most commonly employed for this task (*Delwart, 2007*; *Mokili, Rohwer & Dutilh, 2012*).

In this study, we examined if the data generated in eukaryotic sequencing projects can be used to identify microbiome taxa and harness genomic data. Previously, this approach was used to recover genomes of heritable microbes that occur in high densities in many arthropod species, and are therefore prone to be retrieved in arthropod sequencing projects. For example, the genomes of multiple *Wolbachia* strains were discovered in *Drosophila* sequencing data, revealing novel *Wolbachia* diversity and patterns of *Wolbachia* evolution (*Salzberg et al., 2005*; *Richardson et al., 2012*; *Siozios, Cestaro & Kaur, 2013*). Furthermore, the software packages such as Glimmer (*Delcher et al., 2007*) and Blobtools/Blobology (*Kumar et al., 2013*) were specifically designed to facilitate and automated the extraction of symbiont genomes from NGS data, and subsequent studies have demonstrated their

efficiency in this task (*Blow et al., 2016*; *Denver et al., 2016*; *Fierst et al., 2017*). In bee sequencing projects, (partial) symbiont genomes were previously detected in bumble bees (*Martinson et al., 2014*), a mason bee (*Gerth et al., 2014*), and the *Apis mellifera* genome (*Cox-Foster et al., 2007*).

Here, we systematically examine a large number of short reads of honey bee (*Apis* sp.) sequencing projects to investigate whether this archived data can be used to retrieve a wider set of microbial associates, including pathogens and gut symbionts. We focus on honey bees because: (1) there is a large number of short read sequencing projects targeting *Apis*; (2) the components of healthy and unhealthy *Apis* microbiomes are well established (*Evans & Schwarz, 2011*; *Kwong & Moran, 2016*); (3) managed populations of the economically important honey bees have been in decline worldwide (*Neumann & Carreck, 2010*), and it was hypothesized that certain bacteria and viruses are key players in this decline (*Cox-Foster et al., 2007*). Thus, any novel genomic data on honey bee symbionts may directly contribute to our understanding of bee disease.

To identify 'contaminants', we here use 'bait' sequences of symbionts and pathogens to screen a large number of short read libraries from *Apis* sequencing projects. We demonstrate that the libraries contain non-target sequences from many sources, some of which reflect the natural honey bee microbiome and virome. We further show that highly covered, and possibly novel symbiont genomes can be retrieved from this contamination. Our study highlights the value of database sequences for exploratory symbiont screens and argues against neglecting the filtered 'contaminants' in sequencing projects.

## MATERIALS & METHODS

### Screening for microbes in Apis short read sequencing libraries

A graphical overview of our screening process can be found in Fig. 1. We compiled two sets of reference sequences to be used as databases for subsequent screens (Fig. 1). First, for 15 common *Apis* associated symbionts and pathogens, we used one short signature sequence each instead of complete genomes, in order to reduce the computational expense of all following steps. These were of slowly evolved ribosomal RNA genes to allow a range of diversity to be recovered through sequence similarity to the bait. Second, for 11 common viral associates, we compiled complete genomes. As all of these were very small (∼10,000 bp), this was unlikely to significantly impact computational run times. Fasta files of compiled database sequences are available under https://github.com/gerthmicha/symbiont-sra.

Next, we searched for honey bee sequencing projects in NCBI's short read archive, using the search term '*Apis*', and created lists of the hits for DNA and RNA sequencing projects, which were subsequently processed separately (Fig. 1). The great majority of projects were from *Apis mellifera* samples, but a few RNA sequencing projects from *Apis cerana* and *Apis florea* were also included. We will from here on refer to all of those samples as '*Apis*'. From the list of DNA sequencing projects, we excluded metagenome and amplicon projects, as these are specifically aimed at microbiome taxa. At the time of the search (October 2015), 18 projects matched these criteria. We downloaded all short read libraries associated with
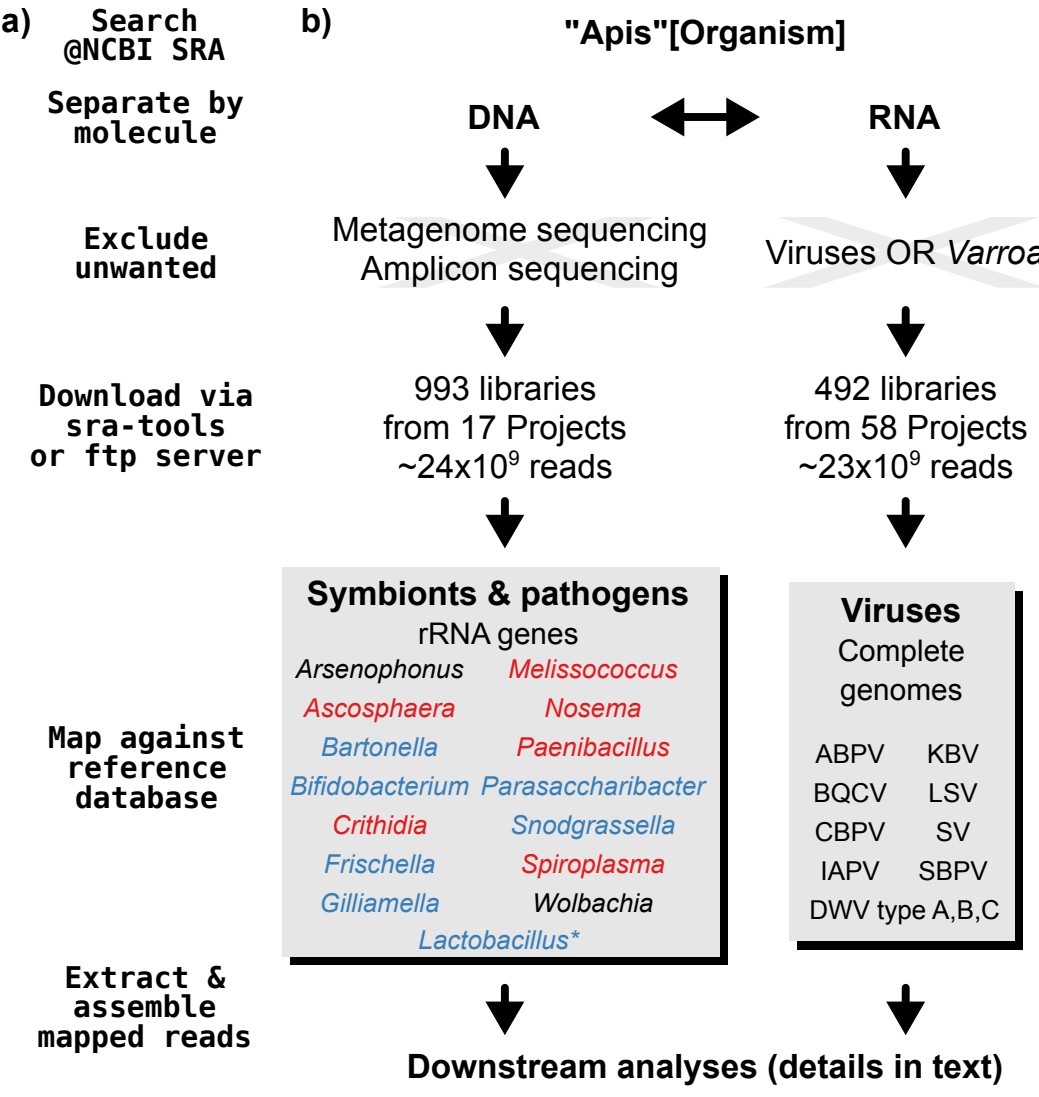

**Figure 1** **Outline of screening procedure employed in this study.** (A) Computational steps involved in screening. (B) Details for each of the steps. For the 'symbionts & pathogens' database, microbial taxa are color-coded as follows: red–pathogens, blue–gut symbionts, black–environmental/opportunistic symbionts. A second, more detailed screen was performed for *Lactobacillus* symbionts only (marked with an asterisk, for details see 'Materials & Methods'). A detailed guide to replicate our analyses can be found under https://github.com/gerthmicha/symbiont-sra.

these projects (993 in total, Table S1) and mapped all reads of each of the libraries to the 'symbionts & pathogens' reference database using NextGenMap version 0.4.12 (*Sedlazeck, Rescheneder & Von Haeseler, 2013*). If at least 1,000 reads of a library were aligned to one or more sequence baits, we extracted the matching reads and assembled them using SPAdes version 3.7 (*Bankevich et al., 2012*). Contigs resulting from this assembly were then subject to taxonomic annotation via BLAST+ (*Camacho et al., 2009*) searches against a local copy of the NCBI 'nt' database, and the Blobtools package (*Kumar et al., 2013*).
For the RNA dataset, we removed any library from projects investigating the effects of viruses or *Varroa* infections, as *Varroa* treatment usually results in viral infections (*Allen & Ball, 1996*). Next, the reads of all remaining RNA sequencing libraries (492 libraries from 58 projects in total, Table S2) were mapped against the viral database as described above, and the number of mapped reads was counted using samtools version 1.4 (*Li et al., 2009*). Mapped reads were then assembled using IDBA-UD (*Peng et al., 2012*) and MEGAHIT version 1.1.1 (*Li et al., 2015*), which were designed to handle highly uneven sequencing depths, as expected from viral data (*Visser et al., 2016*). The 'better' assembly from these two methods was chosen by means of viral contig length comparison, i.e., the assembly that created the single longest contig was kept. This was more appropriate than choosing the highest N50 value, as most libraries contained only a single viral associate. If the two assembly methods produced overlapping contigs, which were identical in the overlap, the contigs were merged. A detailed description of all steps used to screen for microbial associates can be found under https://github.com/gerthmicha/symbiont-sra.

## Assessing *Lactobacillus* diversity in *Apis* DNA sequencing libraries

Since our initial screen of DNA sequencing libraries yielded a high number of hits to various *Lactobacillus* species, we repeated the entire procedure using a database of 620 16S bait sequences from *Lactobacillus* only. These sequences were taken from a previously compiled dataset of Lactobacilli associated with *Apis*, other Hymenoptera, and other *Lactobacillus* sequences retrieved from public databases (*McFrederick et al., 2013*). All hits shorter than 250 bp were discarded, and remaining contigs were combined with the reference sequences. We used SSU-ALIGN version 0.1 (*Nawrocki, 2009*) to align and mask this dataset based on conserved secondary structure. Original and masked alignments are available from https://github.com/gerthmicha/symbiont-sra. A maximum likelihood phylogeny was reconstructed from the complete 16S alignment (740 sequences in total) using IQ-TREE version 1.3.10 (*Nguyen et al., 2015*) with automated model selection and 1,000 ultrafast bootstraps (*Minh, Nguyen & Von Haeseler, 2013*) to assess node support. The resulting tree was visualized using the online tool Evolview (*He et al., 2016*). Furthermore, as an approximate measure for the number of *Lactobacillus* OTUs recovered with our approach, we used the average neighbor clustering algorithm as implemented in mothur version 1.34.4 (*Schloss et al., 2009*).

Although our aim was not to recover all, but only the highly covered symbiont data from honey bee short reads, we wanted to test if our screening approach yields comparable results to more commonly used metagenomic approaches. To this end, we screened the reads of a metagenomic dataset created from the pooled DNA of 150 honeybee worker hindguts (*Engel, Martinson & Moran, 2012*; ~43 M 150 bp paired-end reads, SRA accession: SRR5237156) for *Lactobacillus* in the same way as described above. We found 6 different *Lactobacillus* 16S sequences, all within the Firm-4 and Firm-5 *Lactobacillus* groups (Fig. S1). This was in agreement to the results obtained from taxonomic profiling approaches performed by *Engel, Martinson & Moran (2012)* and thus confirmed the general effectiveness of our approach (Fig. S1).

## Reconstruction of symbiont genomes from *Apis* DNA sequencing libraries

Next, we aimed to validate that whole symbiont genomes can in principle be recovered from *Apis* sequencing projects. To this end, we chose one *Apis mellifera intermissa* sequencing library (SRR1046114, ∼85.5 M 100 bp paired-end reads) that contained 'contamination' from two *Lactobacillus* strains (*Lactobacillus kunkeei* & *Fructobacillus* sp.). We performed a *de novo* assembly using all reads with MEGAHIT version 1.1.1 (*Li et al., 2015*). All resulting contigs of this assembly were taxonomically assigned to either *L. kunkeei*, *Fructobacillus* sp. or 'other' based on BLAST searches, GC distributions, and read coverage. Reads matching to contigs from either *Lactobacillus* strain were then separately re-assembled using SPAdes, and all contigs smaller than 500 bp discarded. Completeness and contamination of the novel draft genomes were assessed based on the presence of conserved marker genes using CheckM version 1.0.6 (*Parks et al., 2015*), and annotation performed with PROKKA version 1.12 (*Seemann, 2014*). The annotated draft genomes are available under under https://github.com/gerthmicha/symbiont-sra. To evaluate the evolutionary relationships of newly assembled genomes in a broader taxonomic context, we assessed their phylogenetic placement. Whole-genome datasets were compiled for both strains (13 *L. kunkeei* genomes, 9 *Fructobacillus* & *Leuconostoc* genomes altogether, Table S3). For each of the datasets, single copy orthologs were identified using OrthoFinder version 0.2.8 (*Emms & Kelly, 2015*). Recombining loci were identified by using the pairwise homoplasy index test (*Bruen, Philippe & Bryant, 2006*), and removed from subsequent analyses (window size = 20 amino acid positions, significance cutoff at 0.05). Using IQ-TREE, we performed maximum likelihood analysis of two final supermatrices (947 loci and 290,774 aa for the *L. kunkeii* dataset, 435 loci and 145,069 positions for the *Fructobacillus*/*Leuconostoc* dataset). Prior to this, best-fitting partitioning schemes and models were selected using the 'greedy' scheme implemented in IQ-TREE (*Lanfear et al., 2012*).

Using the same approach, we assembled and annotated a *Spiroplasma melliferum* genome from the *Apis mellifera* library SRR957082, (∼224.5 M 50 bp single end reads). Phylogenetic analysis was performed based on a dataset of 206 concatenated single copy genes (58,950 amino acid positions) shared among 17 *Spiroplasma* strains (Table S3). Furthermore, to assess synteny, the newly assembled draft genome was ordered against and aligned with other *Spiroplasma melliferum* genomes (one genome each of strains IPMB4A and KC3) using the progressiveMauve algorithm of Mauve development snapshot version 2015-02-13 (*Darling et al. 2010*).

## Assessing viral diversity in *Apis* RNA sequencing libraries

All potential viral contigs resulting from the assembly steps outlined above were blasted against a local copy of NCBI's 'nt' database. As most contigs were annotated as deformed wing virus (DWV), we reconstructed the phylogenetic relationships between the detected DWV strains. We used only contigs where at least half of the DWV genome was present (5,000 bp minimum length), and aligned all sequences using MAFFT. As a reference, we added complete genomic sequences for each of the three types of DWV (A, B or VDV-1–Varroa destructor virus 1, and C) and one for Kakugo virus (KV), which is a
**Table 1** Number of differentially expressed genes per investigated RNA sequencing project.

| BioProject accession number | Number of investigated RNA sequencing libraries with/without viruses | Number of differentially expressed genes[a] |
|---|---|---|
| PRJNA243651 | 9/12 | 7 |
| PRJNA292006 | 3/2 | 5 |
| PRJNA306498 | 3/21 | 5 |
| PRJNA338450 | 17/19 | 10 |

**Notes.**
[a]Out of 20 candidate loci investigated, see Table 2.

variant of DWV type A (*Mordecai et al., 2016*). The phylogeny was estimated via maximum likelihood with IQ-TREE as described above.

## Exploring viral effects on *Apis* gene expression from RNA sequencing libraries

We observed that in some cases, libraries from a single project (i.e., NCBI's BioProject) displayed pronounced differences in viral loads as measured in our screen. We thus hypothesized that these differences should also be detectable through differentially expressed host genes. To test this hypothesis, four projects were investigated (Table 1) in which viral loads were particularly uneven (proportion of viral reads across libraries 0%–37%, 0%–15%, 0%–53%, and 0%–32%, respectively—see Table S4 for details). Several studies have examined differential gene expression of honey bees in response to viral infection, and a set of genes commonly responding to pathogens was recently identified via meta-analyses (*Doublet et al., 2017*). We used the 20 genes that were consistently differentially expressed across multiple studies (first 20 ranks in the category 'differentially-regulated' from the *Doublet et al., 2017* analysis). The relative expression levels of these genes were determined for all libraries of the selected projects by (1) quality trimming of all reads with cutadapt version 1.1.3 (*Martin, 2011*; options: -q 25,25 –minimum-length 50 –pair-filter=both) (2) mapping the reads with NextGenMap (min. identity 99%); (3) counting the reads with samtools; (4) normalizing read counts to reads per kilobase per million mapped reads (RPKM) and log 2 transforming this value, (5) allocating the libraries from each project to either 'virus' or 'virus-free' groups based on the proportion of viral reads in the libraries; (6) testing for differentially expressed genes between these groups using the qCML method and exact tests in the R package 'edgeR' (*Robinson & Smyth, 2007*; *Robinson, McCarthy & Smyth, 2009*; *R Core Team, 2015*). Genes were regarded as differentially expressed for *P*-values $\leq 0.05$.

## RESULTS

Using bait sequences of 15 common *Apis-* associated microbes, we found non-target symbiont sequence data in 105 of the 993 investigated DNA sequencing libraries ($\sim$11%). Assembly of symbiont reads resulted in a total of 782 contigs from these 105 libraries (1–57 contigs per library, Table S5), which were on average 371 bp long (34–1670 bp, Table S5). Taxonomic annotation revealed that these contigs originated from either *Apis-* associated

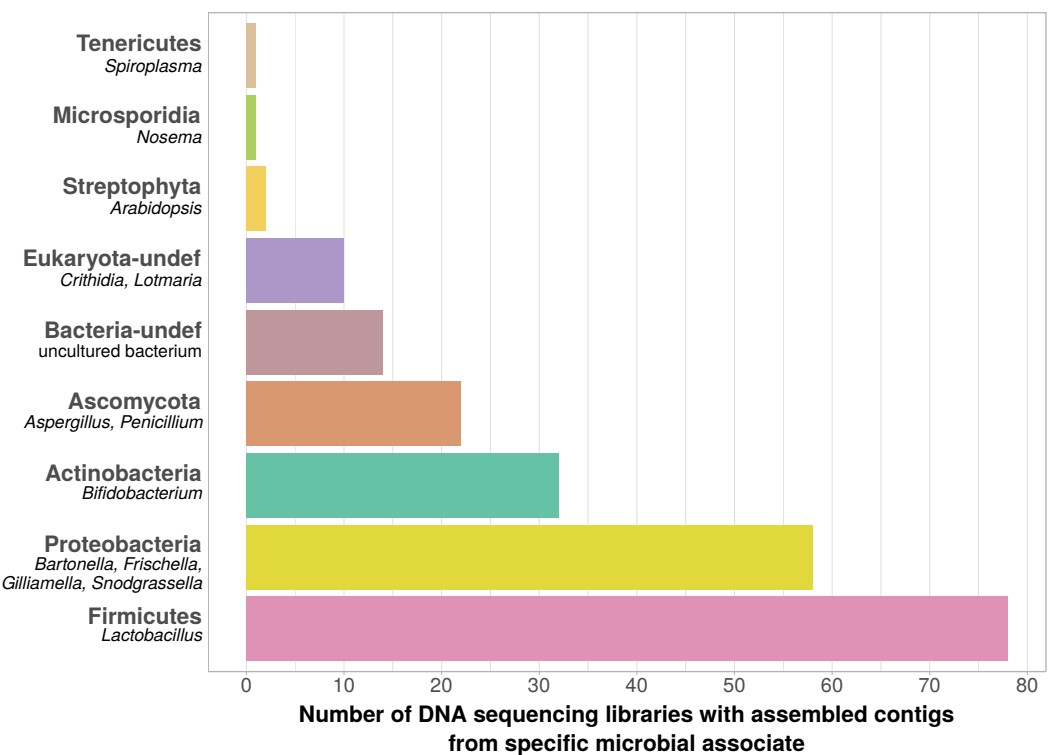

**Figure 2** **Taxonomic annotation of contigs assembled from 'contaminated' *Apis* DNA short read sequencing libraries.** Bar chart shows the frequency of each taxonomic category assigned by best BLAST matches against NCBI's 'nt' database, as the number of libraries in which that taxon was detected among 993 DNA sequencing libraries. Bold categories are 'phyla', as defined in https://www.ncbi.nlm.nih.gov/taxonomy, taxa in italics represent typical genera that were recovered within each phylum. Please note that best blast hits to Arthropods were not included in this plot. See Table S5 for a complete list.

taxa that were targeted with our bait sequences, or from other sources for which there is no current evidence of *Apis* association (Fig. 2). Of the former, we detected most common gut symbionts known to be associated with honey bees (*Kwong & Moran, 2016*), i.e., *Bartonella*, *Bifidobacterium*, *Frischella*, *Gilliamella*, *Lactobacillus*, and *Snodgrassella*, but not *Parasaccharibacter*. Of the pathogens screened for, we detected *Crithidia/Lotmaria* (which are inconsistently annotated in NCBI's nucleotide database and therefore not differentiated here; *Schwarz et al., 2015*), *Nosema*, and *Spiroplasma* (Fig. 2). We did not find evidence for chalkbrood, American foulbrood, or European foulbrood in the investigated DNA libraries (*Ascosphaera*, *Paenibacillus*, and *Melissococcus*, respectively). Sequences from organisms not naturally associated with honey bees included those from fungi (Ascomycota) and plants, that were likely not part of the native microbiome of the sequenced samples. These contaminations were crossed-checked via manual online BLAST searches and were confirmed to represent 'true' hits with high and continuous identities with the respective database sequences.

Because the majority of hits in the DNA sequencing libraries were Lactobacilli, we repeated the screening, this time using only *Lactobacillus* 16S sequences as baits. We found 121 *Lactobacillus* sequences in 40 of the 993 investigated libraries, corresponding to 25

**Table 2  Evidence for differential gene expression of honey bees in response to viral infections across four RNAseq projects (see Table 1).** 20 candidate loci were investigated and differential expression was determined with edgeR (see text for details).

| Locus | Description | Number of projects in which differentially expressed* |
|---|---|---|
| LOC410087 | Uncharacterized LOC410087 | 4 |
| LOC408807 | Uncharacterized LOC408807 | 3 |
| LOC406114 | Alpha-amylase | 2 |
| LOC406142 | Hymenoptaecin | 2 |
| LOC724239 | Kynurenine/alpha-aminoadipate aminotransferase, mitochondrial-like | 2 |
| LOC724367 | Protein lethal(2)essential for life-like | 2 |
| LOC724654 | Cytochrome b5 type B-like | 2 |
| LOC726418 | Flavin-containing monooxygenase FMO GS-OX-like 3-like | 2 |
| Vg | Vitellogenin | 2 |
| Apid1 | Apidaecin 1 | 1 |
| Def2 | Defensin 2 | 1 |
| LOC406144 | Abaecin | 1 |
| LOC725017 | UDP-glycosyltransferase | 1 |
| LOC725725 | Uncharacterized LOC725725 | 1 |
| Melt | Melittin | 1 |
| CYP6AQ1 | Cytochrome P450 6AQ1 | 0 |
| Def1 | Defensin 1 | 0 |
| LOC413908 | Cytochrome P450 6AS12 | 0 |
| LOC552832 | Glycine N-methyltransferase-like | 0 |
| LOC725158 | Peptidoglycan recognition protein S1 | 0 |

OTUs (estimated with mothur using a 5% cutoff). In our phylogenetic analysis based on 16S rRNA sequences, most of the detected strains clustered within *Lactobacillus* groups known to be associated with honey bees (Fig. 3A). Of the recovered sequences not clustering within these lineages, three were found to group with other *Apis-* associated Lactobacilli as sister group to the *Lactobacillus coryniformis* group (Fig. 3A). Online BLAST searches revealed *Fructobacillus* species as closest matches based on 16S rRNA sequence.

Next, we aimed at recovering draft genome sequences of bee-associated Lactobacilli. We chose an *Apis mellifera intermissa* sequencing library from which 16S sequences of both *L. kunkeei* and *Fructobacillus* isolates were detected in our screen. The contigs of a meta-assembly were taxonomically annotated, and reads matching to the respective target taxa were then assembled and annotated separately. For each assembly, we performed a phylogenetic analysis based on all single copy orthologs shared with related genomes (Figs. 3B and 3C), thus confirming the identity of the strains as *L. kunkeei* (Fig. 3B) and *Fructobacillus* (Fig. 3C). Both genomes were highly covered and mostly complete based on the presence of conserved markers (Fig. 3D). Finally, we recovered the genome of a *Spiroplasma melliferum* strain from another *Apis* sequencing library (Fig. 4). In the meta-assembly, *Spiroplasma* and *Apis* contigs could be clearly separated by coverage

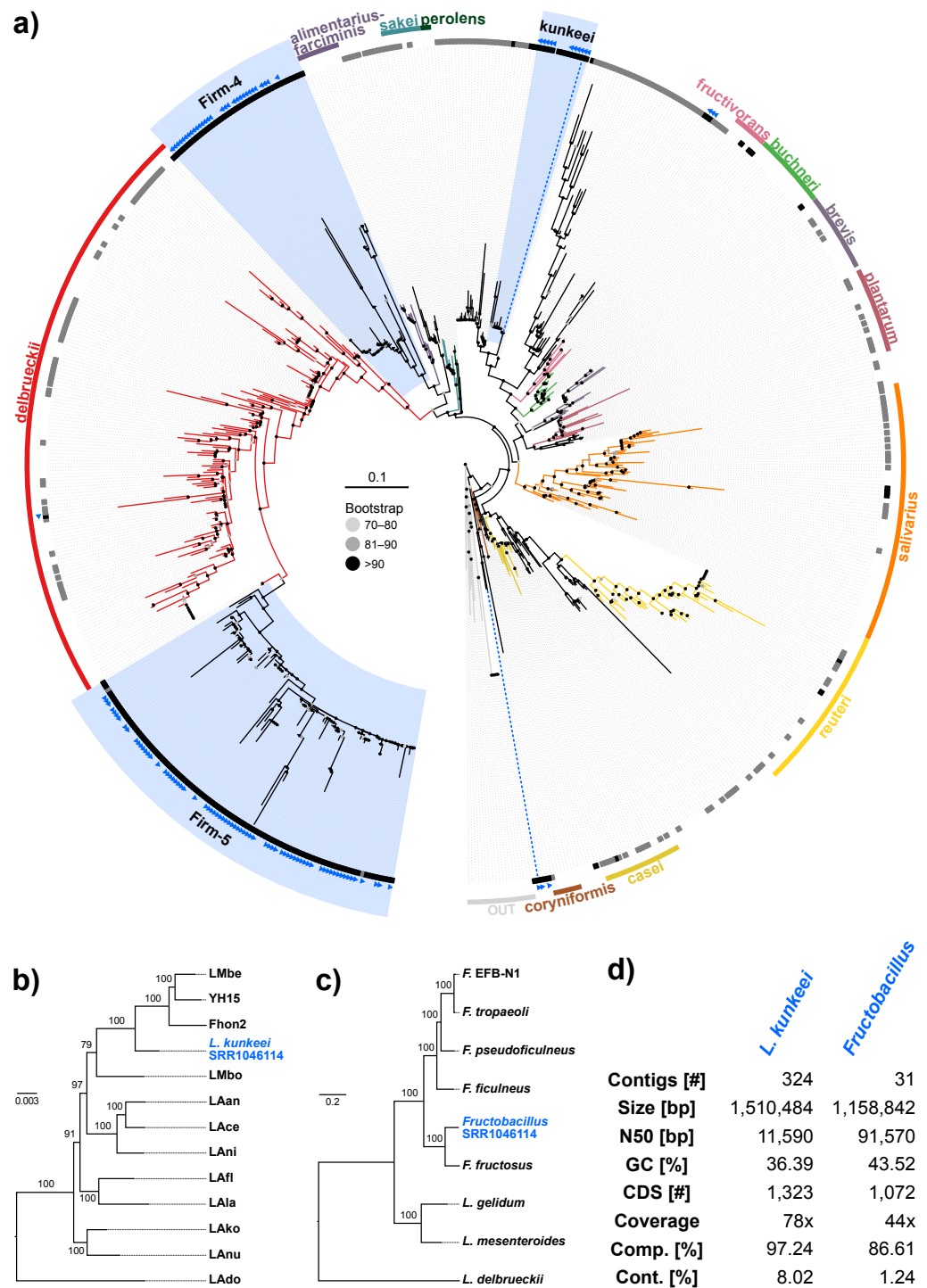

**Figure 3** **'Contamination' from Lactobacilli in *Apis* short read libraries.** (A) Maximum likelihood tree of 720 16S rRNA sequences from Lactobacilli. Branch colors and the color of the outer annotation circle correspond to *Lactobacillus* species groups according to *Felis & Dellaglio (2007)*. Inner circle demarks taxa found Hymenoptera (grey squares) and in corbiculate apids (honey bees and relatives, black squares). *Lactobacillus* sequences recovered in this study from contaminated *Apis* libraries are labeled with blue triangles. The Lactobacilli typically associated with honey bees (continued on next page...)

**Figure 3 (…continued)**
(Firm-4, Firm-5, *L. kunkeei*) are further highlighted with a blue background color. Two dotted blue lines denote the taxa of which whole draft genomes were recovered. See text for details. An interactive version of the tree containing all node labels is available under http://www.evolgenius.info/evolview/#shared/ wZcKHbwJuT. (B) Phylogeny of *Lactobacillus kunkeei* strains based on maximum likelihood analyses of 947 concatenated single copy orthologs (290,774 amino acid positions). Tree is rooted with *Lactobacillus apinorum* Fhon13 (taxon not shown). Strain names correspond to the names used in Tamarit et al. (*2015*; see Table S3). Blue taxon label corresponds to the *L. kunkeei* strain recovered from 'contaminants' in library SRR1046114. Bootstrap values are given on nodes. See Table S3 for sources of genomes. (B) Maximum likelihood tree of *Fructobacillus* (F.) and *Leuconostoc* (L.) species based on 435 concatenated single copy orthologs (145,069 amino acid positions). Tree is rooted with *Lactobacillus delbruecki*. Numbers on nodes correspond to bootstrap values. Again, blue taxon label denotes the *Fructobacillus* genome recovered from the 'contaminated' library SRR1046114. Note that the phylogenetic distance between *Fructobacillus fructosus* and the novel genome is similar to other between-species distances in this tree. See Table S3 for accession numbers of all genomes used for phylogenetic analysis. (D) Assembly statistics for the two novel draft genomes recovered from library SRR1046114. Abbreviations: CDS, coding sequences predicted with PROKKA; Comp. & Cont., completeness and contamination as estimated with CheckM version 1.0.6 (*Parks et al., 2015*) based on the number of conserved marker loci. Phylogenetic affiliations of the two strains are depicted in Figs. 3B and 3C, respectively.

and taxonomic annotations (Fig. 4B). The refined assembly resulted in a highly covered draft genome of *Spiroplasma melliferum*, which is very similar to the two previously sequenced *Spiroplasma melliferum* strains (*Alexeev et al., 2012*; *Lo et al., 2013*), based on shared ortholog clusters, genome organisation, and phylogeny (Figs. 4A, 4C and 4D).

The second part of our screen was focused on viruses in honey bee RNA sequencing libraries. Here, whole viral genomes were used as baits, and viral reads were found in about half of the investigated libraries (236/492 libraries with 500 or more viral reads). Deformed wing virus type A (DWV A) was by far the most frequently found virus, and was present in almost all libraries that contained viral reads. We found evidence for all other tested honey bee viruses as well, albeit at a lower frequency (Fig. 5A). In many cases, the viral reads made up only a negligible proportion of a sequencing library (Fig. 5B). However, in 75 RNA libraries, more than 5%, and up to 54% of the reads were of viral origin, indicating substantial viral loads in the honey bees from which the samples were originally prepared (Fig. 5B). We next investigated the genetic diversity of DWV variants by assembling draft genomes from the viral reads extracted from RNA sequencing libraries. Our phylogenetic analysis showed a clear separation between A and B types of DWV and considerable genetic variation is evident especially in the DWV A samples (Fig. 6). Furthermore, viruses extracted from libraries of the same project were often phylogenetically closely related (Fig. 6), which is not surprising given that samples in one project were commonly from a single geographic location (Table S2).

Viral infections usually result in a host response that should be measurable by means of differentially expressed genes (e.g., immunity-related genes). We tested this prediction using RNA sequencing libraries from four experiments (NCBI's BioProjects) that showed strong variations in viral loads across libraries, and 20 candidate loci that were previously found to be differentially expressed in response to pathogens (*Doublet et al., 2017*; see Table S6). When comparing RNA sequencing libraries with and without viruses, 15 of these genes were differentially expressed in at least one of the investigated experiments

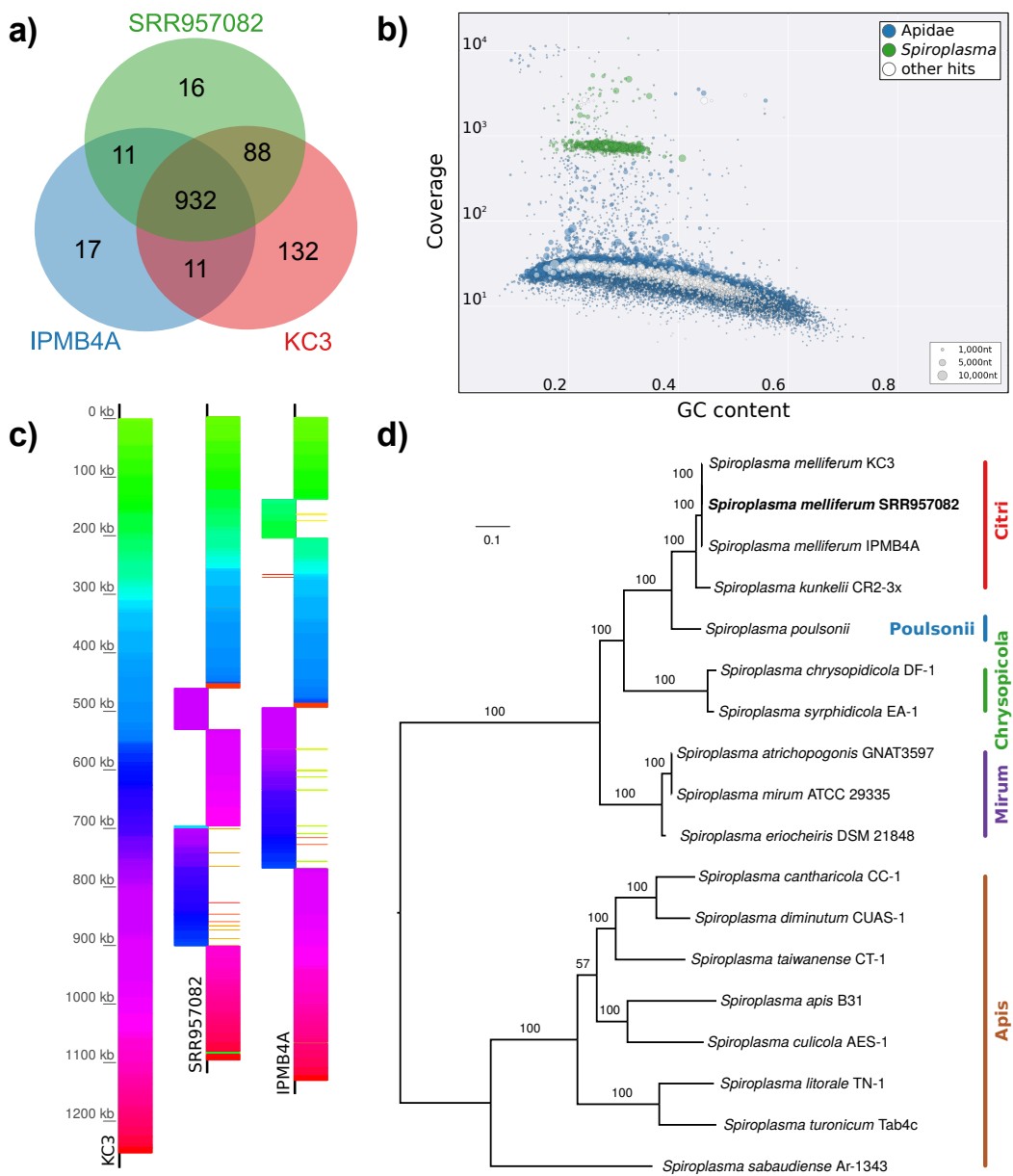

**Figure 4** **Characteristics of *Spiroplasma melliferum*. isolated from a 'contaminated' *Apis mellifera* sequencing library (SRR957082).** (A) Venn diagram illustrating the number of ortholog groups shared between the novel strain and its closest sequenced relatives IBMB4A (*Lo et al., 2013*) and KC3 (*Alexeev et al., 2012*). (B) Taxon-annotated GC-coverage plot of SRR951082 metaassembly created with Blobology. *Spiroplasma* and *Apis* contigs can be differentiated by coverage. (C) Synteny across *Spiroplasma melliferum* genomes. Contigs from assemblies SRR957082 and IPMB4A were ordered against KC3, the most complete of the three *S. melliferum* genomes. Identical colors indicate syntenic blocks of the genomes. (D) Phylogenetic relationships within the genus *Spiroplasma*. Maximum likelihood tree is based on 206 concatenated loci (58,950 amino acid positions), numbers on branches correspond to bootstrap values. *Spiroplasma* groups are highlighted with colors. The taxon label of the novel genome is highlighted in bold. Accession numbers for all taxa are listed in Table S3.

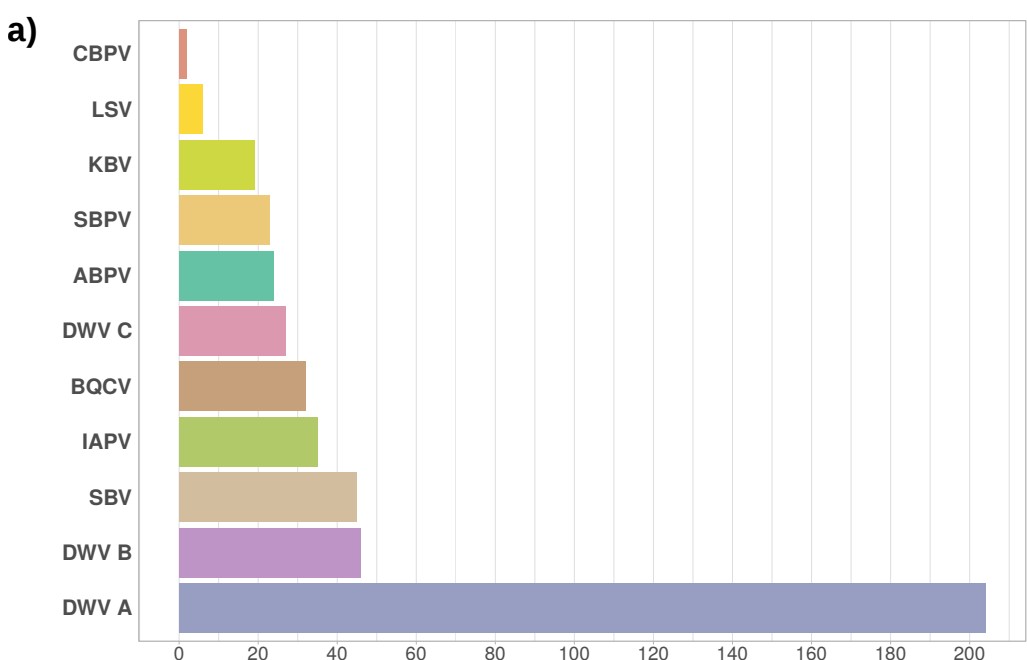

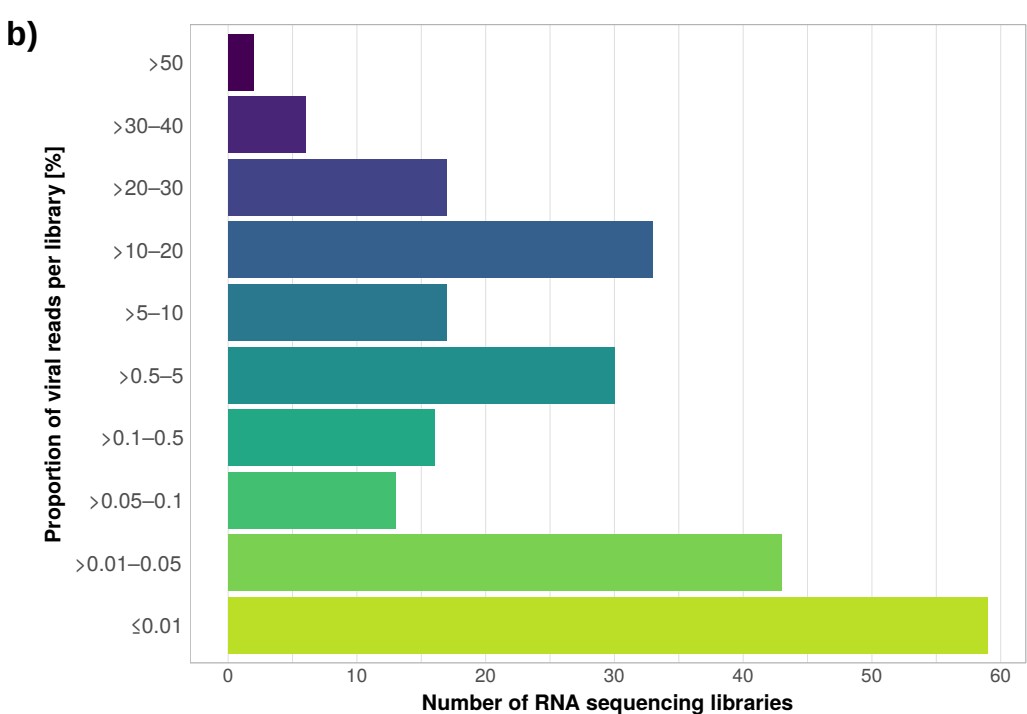

**Figure 5 Distribution of viral reads in *Apis* RNA sequencing libraries.** (A) Bar chart shows how often each of the viruses associated with honey bees was detected with our approach. Results are only shown for libraries with at least 500 reads of viral origin (236 in total). (B) Bar chart shows the distribution of relative proportions of viral reads for these 236 libraries.

**Figure 6** **Phylogeny of DWV sequences extracted from *Apis* RNA sequencing libraries.** Tree is based on 10,291 aligned positions and only sequences with at least 5,000 bp were included. Reference sequences and corresponding accession numbers are highlighted with bold typeface. Colors at nodes indicate NCBI's BioProjects from which the sequences originated.
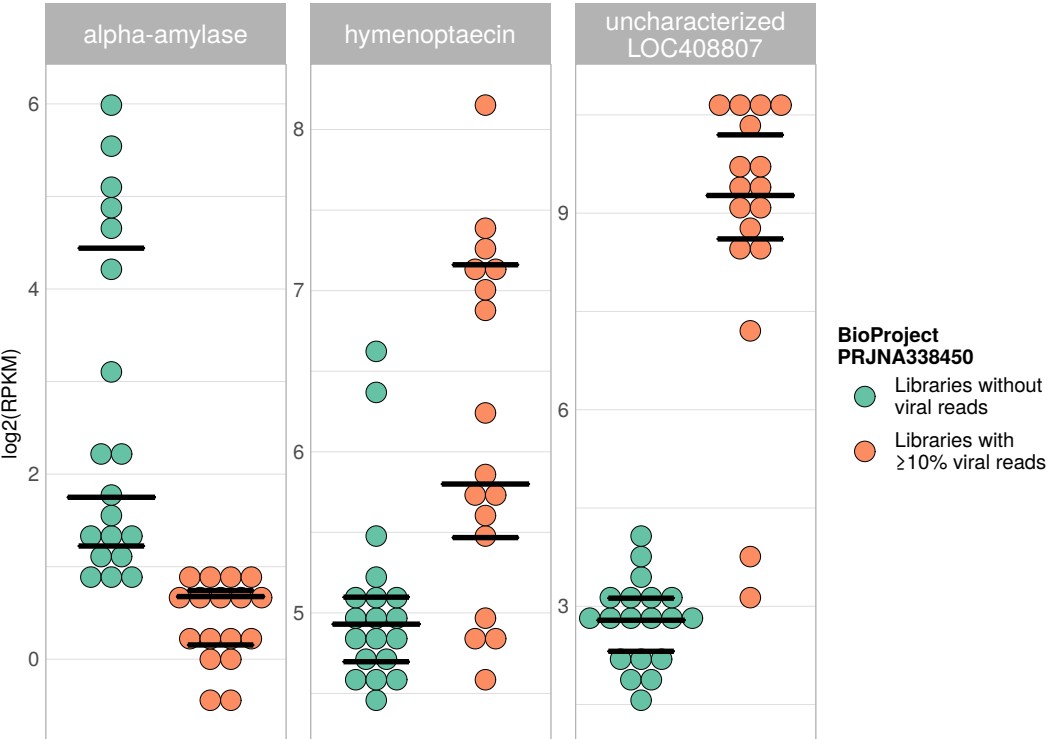

**Figure 7  Example of differentially expressed honey bee genes in response to viral infection.** The relative expression levels of three genes that were determined to be significantly differentially expressed with respect to viral infection status are plotted for one project. Each library is represented by one dot, and lines correspond to 75% quartile, median, and 25% quartile of all samples of a group, respectively. Expression levels for all investigated genes and projects can be found in Figs. S2–S5 and Table S5.

(Table 2), and one gene was found to be differentially expressed in all of them (Table 1). This gene (LOC410087) is not functionally characterized in honey bees, but likely encodes a protein related to heat shock response, which may be activated during viral infections (*Merkling et al., 2015*). Within each experiment, at least five of the investigated genes were differentially expressed between the two groups (Table 1). Relative expression rates of three exemplary genes that were significantly different between libraries with and without viral reads are plotted for one project in Fig. 7. Expression plots for all genes from all investigated projects are shown in Figs. S2–S5.

# DISCUSSION

## *Apis* DNA sequencing libraries can help to reconstruct the honey bee microbiome

We used bait sequences of *Apis* symbionts and pathogens to determine if microbial data can be retrieved from DNA sequencing projects targeting *Apis* (honey bees) and found evidence for the presence of these taxa in 11% of 993 *Apis* short read libraries. This measure of non-target 'contamination' can be considered as conservative, since our approach only reports relatively high levels of contamination (at least 1,000 reads per bait sequence). Our approach revealed that all common gut symbionts of honey bees are also present as

'contamination' in DNA sequencing libraries (Fig. 2). Furthermore, the relative frequency of each of the gut colonizers in focused study roughly corresponds to the frequencies at which we detected them. For example, the microbiome of healthy honey bees is dominated by Lactobacilli (*Kwong & Moran, 2016*), and this is also reflected in our results (Fig. 2, Table S5). Our findings demonstrate that a reasonable understanding of honey bee gut microbiome composition could be gained solely from non-target sequences produced as a by-product of honey bee sequencing projects.

When we next targeted our screen of DNA sequencing libraries only at *Lactobacillus*, our protocol detected 25 taxonomically different *Lactobacillus* strains. Phylogenetic reconstruction of *Lactobacillus* relationships based on 16S rRNA generally reflected the current understanding of this genus' taxonomy (*Felis & Dellaglio, 2007*; *Salvetti & Torriani, 2012*), and revealed that most Lactobacilli known to be associated with honey bees are also present in *Apis* short read libraries. This includes Firm-4 and Firm-5 Lactobacilli, both of which are honey bee hindgut colonizers, and *L. kunkeei*, which is common in nectar and hive material, and sometimes found in honey bee crops (*Kwong & Moran, 2016*). Furthermore, we found *Fructobacillus,* which share an ecological niche with *L. kunkeei,* i.e., they are found in flowers, nectar, and in honey bee guts (*Endo, Futagawa-Endo & Dicks, 2009*; *Endo & Salminen, 2013*). Although not classified as such, recent phylogenomic evidence suggests that *Fructobacillus* (and the closely related *Leuconostoc*) are part of the *Lactobacillus* radiation (*Sun et al., 2015*). Here, we also infer *Fructobacillus* grouping within, rather than outside of *Lactobacillus* (Fig. 2A).

In addition to the gut symbionts, three common honey bee pathogens were detected with our approach: *Nosema*, *Crithidia/Lotmaria*, and *Spiroplasma. Nosema* are microsporidian gut parasites of various honey bee species, and while the sampling of our screen is not representative, this finding corroborates the recognition of *Nosema* as widespread pathogen of honey bee colonies worldwide (*Nixon, 1982*; *Klee et al., 2007*). *Crithidia/Lotmaria* (Trypanosomatidae), another gut pathogen of *Apis* and related bee species (*Schwarz et al., 2015*) was detected at an even higher frequency (Fig. 1, Table S5). We further found *Spiroplasma melliferum* in one of the investigated sequencing libraries. *Spiroplasma* are common symbiotic bacteria of many invertebrates (*Duron et al., 2008*) and have been connected to pathogenicity in honey bees (*Clark, 1977*). The bait sequences of all of these pathogens showed a high coverage in our screen, suggesting that novel genetic variants can be recovered from already available data, or from data that will become available as by-product of future honey bee sequencing projects.

In addition to the targeted microbes, a number of taxa that are likely not part of the natural *Apis* microbiome were detected. For example, we detected *Aspergillus* in several sequencing libraries that originated from museum material, which likely represents post mortem saprophytic growth. We also retrieved hits to plant sequences which might originate from co-amplified and sequenced pollen DNA (Fig. 2). This 'false discovery' illustrates an important caveat in our approach: the differentiation between host-associated microbes and microbes from other sources may not always be possible, and will be particularly difficult for museum specimens. Though not problematic in the examples we present, the situation is likely more complicated for study of host species with a less well-investigated

microbiome, or for symbionts that are very similar to environmental taxa. In these cases, the approach will establish candidates that will then require direct validation.

Finally, as demonstrated previously in other taxa with very similar approaches (*Kumar et al., 2013*), we show that draft genomes of microbial symbionts can be recovered from *Apis* short reads. For example, inspecting the non-target components of just a single *Apis mellifera intermissa* sequencing library produced novel, highly covered, and near complete draft genomes of *Lactobacillus kunkeii* and a *Fructobacillus* strain (Figs. 3B–3D). Although the 16S sequence of the *Fructobacillus* strain best matched *F. fructosus*, our analysis suggests it belongs to a species so far not represented by genomic sequences in public databases, or even a novel species (Fig. 3C). Conceivably, many additional *Lactobacillus* variants could be retrieved from the libraries investigated here, potentially providing a more complete picture of the *Apis* microbiome composition and function. It should be noted that draft genomes reconstructed this way must be regarded as 'population consensus' genomes, as opposed to genomes sequenced from cultured bacterial clones. While these genomes cannot be linked to a bacterial clone, they still provide information of metabolic capacities within the *Apis* microbiome.

In summary, our analyses suggest that 'contamination' from honey bee sequencing projects can help to characterize the honey bee microbiome, and inform about its composition, abundance of specific taxa, and through recovery of genomes, about metabolic capacities. Short read repositories thus provide plenty of biological information on known associates, but also on undescribed ones. If, for example, a novel microbial associate of honey bees is discovered, one could potentially use the available short read data (and associated metadata) to learn about its distribution, prevalence, and genome. Or, alternatively, one could confirm the absence of certain taxa with more confidence. *Kwong et al. (2017)* identified *Bombiscardovia* sp. and *Schmidhempelia* sp. as gut microbes of bumble bees (*Bombus* sp.) that are typically not present honey bees. We also did not find evidence for these taxa in *Apis* with our screening approach and thus provide further support for their absence based on a large sample size.

### Implications of viral reads as common contaminations in *Apis* RNA sequencing libraries

We found viral reads in about half of the 498 investigated RNA libraries, and DWV type A as the most common viral associate, followed by DWV type B (=Varroa destructor virus-1), and all other viruses we screened for (Fig. 5). DWV has a global distribution (*Genersch & Aubert, 2010*) and its transmission is facilitated by *Varroa* mites, which are also globally distributed (*Martin et al., 2012*; *Wilfert et al., 2016*). Our findings are thus in line with these previous observations. The detection of DWV despite our deliberate exclusion of RNA sequencing experiments which investigated *Varroa* or virus treatments can be explained by the fact that, in many cases, DWV infections are asymptomatic (*Lanzi et al., 2006*). The low proportion of DWV reads found in many samples (as a proxy for viral load) also argues for asymptomatic infections in the bees from which the RNA was extracted (Fig. 5). However, we found a number of RNA sequencing libraries in which DWV read numbers made up a very large proportion of the reads (10–54%, Fig. 5). Very

high viral titers are typically found in symptomatic bees (*Chen, Higgins & Feldlaufer, 2005*; *Lanzi et al., 2006*; *Brettell et al., 2017*), and inspection of the metadata associated with RNA libraries showing high proportions of viral reads revealed that many of these were created from heads or brains (out of 58 libraries with >10% viral reads, 25 from heads, 10 from other tissues, 23 undisclosed, Table S4). This is notable because high DWV titres in honey bee heads have been found to correlate with deformed wing phenotypes (*Yue, 2005*; *Zioni, Soroker & Chejanovsky, 2011*). This suggests that for these specific experiments, RNA may have been extracted from DWV infected bees despite obvious symptoms (i.e, crippled wings).

*Varroa* mites and DWV are considered one of the main factors driving the decline in managed honey bee populations (*VanEngelsdorp et al., 2009*; *Conte, Ellis & Ritter, 2010*), and molecular data have been used to trace the global spread of these pathogens (*Martin et al., 2012*; *Wilfert et al., 2016*). We here showed that almost complete DWV draft genomes can be extracted from short read data, revealing considerable genetic diversity between the strains (Fig. 6). Archived RNA sequencing reads of honey bees are thus a valuable, largely untapped resource for genomic data of viruses. As described above for symbionts in DNA sequencing libraries, these data inform about host association, distribution, and genetic diversity of viruses.

Furthermore, the detection of large amounts of viral reads in honey bee RNA sequencing libraries has other, more direct implications. Differential gene expression analysis was previously used to study e.g., development, learning, caste differences, and other behavioral traits in honey bees (*Whitfield, 2003*; *Liu et al., 2011*; *Liang et al., 2012*; *Cameron, Duncan & Dearden, 2013*; *Naeger & Robinson, 2016*). It has further been established that viral infections lead to transcriptomic responses in honey bees, some of which are conserved and universal for all pathogens, while others are specific with respect to the pathogen present (summarized in *Doublet et al., 2017*). Following from this, the presence of viruses alters transcriptomic profiles of honey bees and therefore incorporated into analyses of differential gene expression, e.g., by adding viral load as one factor in the interpretation. In most experiments analysed here, the presence of viruses was likely not a problem, as viral reads were uniformly low across all samples. However, we did identify some examples in which uneven viral loads very likely had an effect on transcriptomic profiles (Tables 1 and 2, Fig. 7 & Figs. S2–S5). In these examples, the presence of viruses, if unaccounted for, potentially leads to erroneous interpretations of the RNAseq data. Consequently, viruses should be screened for and identified in any RNA sequencing experiment investigating differential gene expression.

## CONCLUSION

The biological properties of an individual are a composite of the functions encoded in their genome and that of microbial associates, both symbionts and pathogens. Here we revisited published short read data from *Apis* spp. sequencing projects to investigate if these give insight into the wider set of associates that are commonly disgarded as 'contaminants'. We found that a large variety of distinct *Apis*-associated microbial symbionts and pathogens

can be detected as 'contamination' in these data. Further, due to the large depths of today's sequencing projects, the genomes of some microbial associates (which are typically much smaller than the target genomes) can often be recovered in high quality. Honey bees have a comparatively simple microbiota (*Kwong & Moran, 2016*) and are thus considered suitable models for microbiome-animal interactions and evolution (*Engel et al., 2016*). Their enormous economic importance (*Calderone, 2012*) has driven a large number of honey bee sequencing projects. Our examination of the output of these projects suggests that substantial amounts of genomic information on bee-associated microbes are included in these data. While genomes gained from contaminated bee samples cannot and should not replace focused microbiological and metagenomic investigations, they might still improve our understanding of honey bee microbiome composition and functioning.

Although our study was focused on *Apis*, it is conceivable that the amount of non-target 'contamination' is similar for other sequencing projects. As a best practice, and potentially rewarding research avenue in any sequencing project, we therefore suggest that all non-target taxa should be identified, and their genomes assembled, annotated, and published alongside the target genome. This requires less effort than it may seem, as de-contamination is already a standard post-processing step. Instead of discarding the contaminated reads, they can be processed with one of many available software solutions that automate the process of identifying and assembling genomes from metagenomes (*Oulas et al., 2015*), thus minimizing the additional workload. Not only would this provide the community with valuable genomic data of symbionts from known host taxa, but it can additionally be argued that this is the most sensible thing to do from a biological point of view. Evidence is mounting that symbiotic microbes influence almost all aspects of their host's biology (*Douglas, 2014*; *Bordenstein & Theis, 2015*). Taking into account the total genomic information recovered in sequencing projects may therefore provide a more complete picture of the target organism's biology.

## ACKNOWLEDGEMENTS

The authors would like to thank Dr Seth Barribeau for comments on an earlier version of this manuscript, and anonymous reviewers for constructive feedback. We further thank Dr Philipp Engel and Olivier Emery for providing access to short read data.

### Funding

This work was supported by the European Commission through H2020 funding in the form of an EMBO long term fellowship (ALTF 48-2015, LTFCOFUND2013, GA-2013-609409) and a Marie Curie Fellowship (H2020-MSCA-IF-2015, 703379) to MG. The funders had no role in study design, data collection and analysis, decision to publish, or preparation of the manuscript.

### Grant Disclosures

The following grant information was disclosed by the authors:

EMBO long term fellowship: ALTF 48-2015, LTFCOFUND2013, GA-2013-609409.
Marie Curie Fellowship: H2020-MSCA-IF-2015, 703379.

## Competing Interests

The authors declare there are no competing interests.

## Author Contributions

- Michael Gerth conceived and designed the experiments, performed the experiments, analyzed the data, contributed reagents/materials/analysis tools, wrote the paper, prepared figures and/or tables, reviewed drafts of the paper.
- Gregory D.D. Hurst conceived and designed the experiments, contributed reagents/materials/analysis tools, wrote the paper, reviewed drafts of the paper.

## Data Availability

Github: https://github.com/gerthmicha/symbiont-sra.

## Supplemental Information

Supplemental information for this article can be found online at http://dx.doi.org/10.7717/peerj.3529#supplemental-information.

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
