# Peer review of "Short reads from honey bee (Apis sp.) sequencing projects reflect microbial associate diversity"

_PeerJ, doi:10.7717/peerj.3529_

## Round 0.1 · original submission · Major Revisions

· Academic Editor

Major Revisions

This manuscript has now received two reviews from people with expertise in bee genomics and bioinformatics. Both Reviewers thought the paper was interesting and I agree that assembly and characterization of co-occurring microbial genomes is an approach that yields potentially very interesting results. Nonetheless, while both Reviewers were supportive of publication, they had suggestions with the presentation that need to be addressed. Some of these caveats potentially include additional analyses that may be beyond minor revisions, but generally I suspect they should be relatively easy to address and incorporate. I look forward to reading your revision.

Reviewer 1 ·

Basic reporting

No comment.

Experimental design

No comment.

Validity of the findings

No comment.

Additional comments

This paper by Gerth and Hurst describes analyzing almost 1000 short read libraries from honey bee sequencing projects to identify and assemble the genomes of co-occurring microbes. The paper seems scientifically sound, and provides sufficient, novel results for publication. I do not have any major reservations about this study, but I strongly suggest the authors address the following issues to clear up some confusing aspects of the paper:

How and why certain taxa were chosen for screening (Table S1)? In my opinion, this table should be part of the main paper, because what is used as bait will undoubtedly bias the results obtained, and hence should be clearly presented. It is also unclear why the known bee gut bacterial associates (Gilliamella, Bartonella, Snodgrassella, Frischella, Bifidobacterium, Lactobacillus) were not included as part of this list of initially screened candidates. Almost certainly the authors will retrieve many reads matching these bacteria, which may or may not be interesting for them; nonetheless, an explanation is required.

It is also unclear what bee species were used: the authors refer to “Apis sp.”, and “honey bees”, but this group encompasses 9 different bee species, which may have different gut associates. It would be good to note somewhere (Table S2?) if these short read libraries come from Apis mellifera, Apis cerana, Apis florea, or other Apis species. This should also be mentioned in the main text.

Although almost 1000 sequencing libraries were screened, how many unique Apis sequencing projects did these correspond to? If multiple libraries were generated from the same individual(s), that would skew the results, and the outcomes (e.g., Fig. 1) should be reinterpreted in this new light.

The third lactobacillus cluster of reads retrieved by the authors in Fig. 2a and S1; could this be the mysterious “Firm-3” found in some previous papers (McFrederick et al. 2013, AEM)?

Finally, although the assembly of microbial sequences from heterogeneous, host-dominated datasets is a respectable technical feat, I would have liked to see more discussion of the biological relevance of the results. For instance, did the authors find anything interesting in the new Lacto/Spiroplasma genomes? Did they find particular associations of the presence of certain microbial taxa with certain samples/hosts?

Minor comments:

Line 106: “short” should be “shorter”
Line 135: explain briefly what metrics CheckM uses to check for genome completeness.
Figure 3c: explain the relevance of the color scheme of the genome representations (if any).
Figures 2a, S1: spell out full species names of the Lactobacillus clades. The abbreviations are difficult to immediately grasp, and there is enough space in the figures to have full names.

Reviewer 2 ·

Basic reporting

Genome sequencing projects provide an abundance of information about the target organism’s genetic composition; however, genome projects often include microscopic organisms that are associated with the host. Exploring the co-sequenced microbial DNA can give insights into the genomes of pathogens, intracellular symbionts, or common host-associated microbes (e.g. gut microbiota). This manuscript reads well and it has a clear message. The surveys performed were well thought out. Overall the structure of the manuscript is very nice, and the figures, supplementary, and raw data materials are well organized.

(1) That being said, I have a major issue with how this study is presented. The authors do not mention or cite much of the background literature on which this work is based. Many of the most intellectually related publications have been excluded.
The program Blobtools is utilized in the methods section (Line: 99); however, the authors do not comment on the reason behind creating Blobtools. This program was directly created to remove bacterial contamination in eukaryotic genome sequencing projects (Kumar et al. 2013 Frontiers in Genetics). Further, in the original manuscript Blobtools was used to directly identify bacterial symbionts in nematode hosts and has been used by others to find more symbionts (Brown et al. 2015 GBE).
Another method that has been used to identify host-associated microbes in genome projects is Glimmer (Delcher et al. 2007 Bioinformatics). And in a related area, searches for horizontally transferred genes in genomes has analyzed genomes to sort microbial contamination from true HGTs (Dunning Hotopp et al. 2007 Science; and many other HGT publications).
Finally, bacterial genomes have previously been found in genome sequencing projects of bees. The Apis mellifera genome sequence information was screened for bacterial ‘contamination’ in Cox-Foster et al. (2007 Science), which found bacterial sequences that corresponded to the common gut bacteria. Additionally, an entire bacterial symbiont genome was sequenced as a by-product of the bumble bee (Bombus impatiens) genome project (Martinson et al. 2014 AEM).

(2) Another omission that should be added to the manuscript is an overview of the SRA Apis files that were screened. Can you provide a list of species/subspecies of Apis were screened? The library SRR1046114 was used to assemble the Lactobacillus kunkeei and Fructobacillus sp. Genomes – what Apis species was this? Were certain pathogens more prevalent in different Apis species/subspecies?

(3) My final (larger) issue is with the use of the terms holobiont and hologenome. These terms are not necessary in your study and are counter to your goals. A holobiont is specifically defined as – host and microbiota are the unit of selection (Brucker & Bordenstein 2013 Science). However, in your genomic survey you find unspecific microbes, common honeybee gut bacteria, and honeybee pathogens. Pathogens are not under the same selective pressures as the host, and thus this is not a survey of the holobiont. I might suggest that throughout your manuscript you simply use “the host and microbiota” or “symbiome” as in Douglas & Werren (2016).

Line Comments
Line: 58-59
Do you mean – genome sequences are less resolved because they are assembled from population of microbes?

Line: 60-62
What do you mean by “inform about the composition of the wider ‘holobiont’”? Do you mean the population-level microbiome diversity? Or is this something specific to the holobiont theory?

Line 84:
There is a project that has assembled many of the genomes of Apis-associated organisms. I thought this might interest you. HoloBee project (https://data.nal.usda.gov/dataset/holobee-database-v20161).

Line: 158-170
Were there only 254 contigs identified as associated organisms (Table S4)? Are the Table S4 contigs just representative contigs for all the taxonomic clusters of DNA sequence that were found in each SRA library? I think it would be good to at least mention a bit more about the contamination you found – the total number of contigs/library, and the sequence length of those contigs.

Line 228-230
This is a good point and it will make careful screening important.

Figure 1 is a bit confusing. If I am interpreting this figure correctly, it illustrates the results from searches of Apis short read libraries. Why do only ~13 seqeuncing libraries contain contigs identified in the taxonomic category “Arthropoda”? In the Fig. 1 legend I suggest a change the wording: … in which that taxon was detected among the 993 SRA libraries.

Table S4
This file is very difficult to interpret, but I think with some fairly simple additions it will be a nice supporting table. It would help if you could add information for each of the SRA libraries (e.g. Apis species, sequencing group, date sequenced, etc.). Also, for the contigs that are categorized as “Arthropoda” some of the information seems mis-sorted – for example see excel line 174. Apis florea is not in the family Membracidae. Has this information been sorted incorrectly?

Experimental design

The experimental design is well described and thought through and I think that it clearly shows that genome sequencing projects can be a jackpot of information about host-associated microbes. However, you only explore the microbial ‘contamination’ of Apis. Would it be good to screen another genome project with your method? Maybe just mention that this could be an avenue of exploration in invertebrate genome projects?

Validity of the findings

The findings are robust and the conclusions are well stated.

---

## Round 0.2 · accepted · Accept

· Academic Editor

Accept

The authors have done an excellent job addressing the previous suggestions for edits and, as a result, the revised manuscript reads really well and will be of broad interest to readers. Indeed, the short read libraries from the honey bee sequencing projects reveals some really interesting outcome. Thank you for sending your work to PeerJ.